# Targeting the RNA Polymerase I Transcription for Cancer Therapy Comes of Age

**DOI:** 10.3390/cells9020266

**Published:** 2020-01-21

**Authors:** Rita Ferreira, John S. Schneekloth, Konstantin I. Panov, Katherine M. Hannan, Ross D. Hannan

**Affiliations:** 1ACRF Department of Cancer Biology and Therapeutics, The John Curtin School of Medical Research, Australian National University, Acton 2601, NSW, Australia; k.panov@qub.ac.uk (K.I.P.); kate.hannan@anu.edu.au (K.M.H.); ross.hannan@anu.edu.au (R.D.H.); 2Chemical Biology Laboratory, Center for Cancer Research, National Cancer Institute, Frederick, MD 21702, USA; jay.schneekloth@nih.gov; 3CCRCB and School of Biological Sciences, Queen’s University Belfast Medical Biology Centre, School of Biological Sciences, Queen’s University Belfast, Belfast BT9 7BL, UK; 4Department of Biochemistry and Molecular Biology, University of Melbourne, Parkville, VIC 3010, Australia; 5Peter MacCallum Cancer Centre, 305 Grattan St, Melbourne, VIC 3000, Australia; 6Department of Biochemistry and Molecular Biology, Monash University, Clayton, VIC 3800, Australia; 7School of Biomedical Sciences, University of Queensland, Brisbane, QLD 4072, Australia

**Keywords:** cancer therapy, RNA polymerase I transcription, ribosome biogenesis, CX-5461

## Abstract

Transcription of the ribosomal RNA genes (rDNA) that encode the three largest ribosomal RNAs (rRNA), is mediated by RNA Polymerase I (Pol I) and is a key regulatory step for ribosomal biogenesis. Although it has been reported over a century ago that the number and size of nucleoli, the site of ribosome biogenesis, are increased in cancer cells, the significance of this observation for cancer etiology was not understood. The realization that the increase in rRNA expression has an active role in cancer progression, not only through increased protein synthesis and thus proliferative capacity but also through control of cellular check points and chromatin structure, has opened up new therapeutic avenues for the treatment of cancer through direct targeting of Pol I transcription. In this review, we discuss the rational of targeting Pol I transcription for the treatment of cancer; review the current cancer therapeutics that target Pol I transcription and discuss the development of novel Pol I-specific inhibitors, their therapeutic potential, challenges and future prospects.

## 1. Why Target RNA Polymerase I Transcription?

In “western countries” cancer is now responsible for the majority of disease related deaths each year [1]. A significant amount of research has been conducted worldwide over the last 5 decades resulting in not only a greater understanding of this disease, but also the development of a range of novel therapies including small molecules, antibodies and immunotherapies. However, cancer is a heterogenic collection of diseases, affecting different tissue and cell types and thus the response to a given cancer treatment is also highly variable [2,3,4]. The advent of precision medicine, by targeting the genetic mutations driving individual cancers has ushered in a new era promising higher selectivity with decreased toxicity as only the mutation affected cells are targeted. However, even this approach has limitations as the number of known driver genes far outweigh the available therapies to target them, meaning most mutations are currently unactionable and treatments still heavily rely on more standard approaches such as chemotherapy. While immunotherapies are delivering remarkable results, not all tumours (<20%) are immune responsive [5], and understanding how to immune-sensitise tumours is an ongoing area of investigation. In response to this, a third approach, that combines the targetedness of personalised therapy with theoretical pan-efficacy, is to selectively target a biological process common to most, if not all, cancers or in other words develop “impersonalised precision medicine”. The therapeutic window is achieved by virtue of tumour cell having increased sensitivity to perturbation of certain essential biological process. Therefore, efficacy is not reliant on tumour cells having mutations in the pathways being targeted. This review focuses on a new class of drugs that fall into this latter category, the targeting of ribosome biogenesis (RiBi).

The transformation of normal cells into cancer cells requires the gradual acquisition of certain characteristics, coined “the hallmarks of cancer” [6,7]. These include self-sufficiency in growth signals, insensitivity to antigrowth signals, evasion of apoptosis, limitless replicative potential, sustained angiogenesis, tissue invasion and metastasis capability [7], deregulated metabolism and immune system evasion [6]. Dysregulation of one biological process in cancer cells that is associated with the two distinct, but coupled processes, cellular growth (size) and division [8], is RiBi, the process of producing ribosomes—the machinery responsible for the translation of messenger RNA (mRNA) into proteins. Cell growth and proliferation are separate processes, as illustrated in the case of cardiac myocyte hypertrophy where these post-mitotic cells cannot divide but with stimulation of RiBi they increase in size [9].

RiBi takes place in sub-nuclear domains termed nucleoli that have long been linked to cancer with the enlargement and increase in the number of nucleoli per cell being used for over a century as a marker of malignancy [10]. More contemporary studies have identified that the increase in number and size of nucleoli is due to the hyperactivation of RNA polymerase I-dependent transcription of ribosomal RNA genes (rDNA) that generate the ribosomal RNAs (rRNAs), the nucleic acid backbone of the ribosomes (reviewed by Drygin et al. [11] and Montanaro et al. [12]). Until recently, the role of elevated RiBi in tumorigenesis was believed to be due to the increased demand of proteins for growth and cell division by the tumour cells [13]. However, research over the last 10–15 years have identified non-canonical roles for rRNA synthesis and the nucleolus suggesting that RiBi may play a more extensive role in both cell homeostasis and malignancy than previously appreciated [14,15,16,17,18].

### 1.1. Ribosome Biogenesis

The 80S ribosomes are composed of two subunits: small subunit (40S) that binds and scans mRNA [19] and the large subunit (60S) responsible for peptide bond formation [20]. Both subunits are composed of an rRNA backbone (40S contains 18S rRNA while 60S is composed of 5S, 5.8S and 28S rRNAs) and a large number of ribosomal proteins (RP). The 18S, 5.8S and 28S rRNA are generated by processing of the 47S pre-rRNA transcribed by RNA polymerase I (Pol I), the 5S rRNA gene by RNA Polymerase III (Pol III) and the multiple RP genes by RNA polymerase II (Pol II).

Human cells contain over 200 copies per haploid genome of the loci encoding for 18S, 5.8S and 28S rDNA, termed rDNA repeats. Each repeat is comprised of the *cis*-regulatory elements and the coding region of the 47S precursor RNA transcript, that is processed to give rise to 18S, 5.8S and 18S rRNAs, and a long intergenic spacer (Figure 1). In humans, these repeats are clustered tandemly in the short arms of the five acrocentric chromosomes, regions usually denominated Nucleolar Organiser Regions (NORs). In the G1 phase of the cell cycle these NORs aggregate in the nucleolus, a subnuclear membraneless domain that facilitates the transcription of the rRNAs by Pol I and their subsequent processing. During mitosis, nucleoli disassemble before being regenerated in early G1 phase.

Given the importance of ribosome biogenesis for dictating cellular growth capacity, transcription of 47S rRNA by Pol I is a highly regulated process, coupled directly with a variety of cellular cues including growth factors and nutrients [21], cell cycle [22] and stress [23,24,25]. Central to the regulation of rDNA transcription is the Pol I complex and a number of other auxiliary proteins whose function is predominantly the transcription of the ribosomal genes which certain exceptions [26]. Mammalian Pol I is a multi-subunit protein complex; with some subunits shared between all RNA Polymerases, such as POLR1C, and POLR1D which are shared with Pol III, while others are Pol I-specific. These include POLR1A (RPA1, A194), POLR1B (RPA2, A127), POLR1E (PAF53), POLR1F (hRPA43), POLR1G (CAST, PAF49) and POLR1H (hRPA12) [27,28,29]. Several of these Pol I-specific subunits interact with other Pol I-specific factors like the upstream binding factor (UBF) (POLR1F and G), the transcription factor RRN3 (POLR1F) and the SL-1 complex (POLR1G) and are important for the formation of the Pre-Initiation Complex (PIC) at the rDNA promoter.

While the Pol I complex in the absence of accessory factors, known as Pol Iα, can initiate transcription non-specifically from DNA templates, high levels of accurate transcription from the rDNA promoter requires an initiation-ready form of Pol I, termed Pol Iβ. This is achieved with the binding of the transcription factor RRN3 to Pol Iα [30]. RRN3 allows the interaction of the Pol I with the promoter Selectivity Factor 1 (SL-1) [31], a complex containing the TATA binding protein (TBP) and five TBP-associated factors (TAF1A, TAF1B, TAF1C, TAF1D and TAF12) [28] that recruits the PIC to the rDNA promoter. Another important factor for rDNA transcription is UBF (also known as UBTF), a sequence-nonspecific transcription factor, that binds the rRNA regulatory regions [32] leading to the displacement of histone H1 and maintaining an open chromatin configuration permissive for binding of additional transcription factors [33,34,35]. UBF and SL-1 function in a cooperative way to ensure high levels of rDNA transcription, with SL-1 stabilizing the binding of UBF to the promoter [36]. UBF also plays an important role in promoter escape [37] and in regulation of elongation [38]. Several other factors have been shown to associate with the PIC and facilitate transcription initiation. One such factor is Topoisomerase IIα that binds to RRN3 and facilitates de novo PIC formation by removing DNA supercoils at the rDNA promoter [39].

Within the nucleolus, the 47S transcript is processed into mature 18S, 5.8S and 28S rRNA that associate with the 5S rRNA and the RPs, and the resultant 40S and 60S are subsequently exported from the nucleolus to the cytoplasm where maturation is completed generating the functional ribosome [40].

### 1.2. Nucleolar Surveillance Pathway

It is now clear that, in addition to its role in RiBi, the nucleolus also has evolved a number of functions not strictly linked to ribosome production which we term here “non-canonical” ribosome functions. One such non-canonical function is the ability of the nucleolus to sense perturbations to RiBi and initiate a process called the “nucleolar stress response“ (NSR) or “nucleolar surveillance pathway” (NSP) to trigger a variety of downstream effects including cell cycle arrest and apoptosis. The best studied NSP mechanism leads to activation of the p53 (TP53) pathway. During times of steady state rRNA transcription MDM2, an E3 ubiquitin ligase, ubiquitinates p53, priming it for proteasome degradation. When a key stage of RiBi, rRNA synthesis, is acutely disrupted through any one of a variety of mechanisms (e.g., DNA damage, mutations in Pol I machinery components, drug’s action), “free” Ribosomal Proteins (RP) RPL5 and RPL11 (as part of the 5S-RNP particle) which are no longer being incorporated into the ribosome, bind and sequester MDM2 thus preventing it from ubiquitinating p53. This in turn results in the rapid accumulation of p53 protein which transcriptionally regulates target genes involved in cell cycle arrest, apoptosis and genome stability (Figure 2). Additionally, p53 independent NSP’s have been described mediated by proteins such as E2F-1 [41], or Retinoblastoma (RB) (for a detailed review see [18,42,43]).

### 1.3. Targeting Pol I Transcription for Therapeutic Benefit

RiBi upregulation, modulated by oncogenes like MYC, is essential for the survival and proliferation of cancer cells [44]. The observation that perturbations in RiBi lead to activation of the p53 pathway, causing cell death, cell cycle arrest or senescence, renders RiBi a prime candidate for targeted therapies to treat cancer. RiBi is regulated at multiple steps, including transcription and processing of rRNAs, synthesis of RP and ribosomal assembly, all of which are possible points for therapeutic inhibition. Targeting RiBi through inhibition of Pol I transcription in particular has several benefits for therapy: 1) Pol I is a highly selective process since Pol I transcribes a single pre-RNA transcript, the 47S rRNA. This has the potential to avoid side effects derived from drugs that target inhibition of Pol II-regulated genes by a variety of mechanisms (for example BRD4 [45] and CDK9 [46] inhibitors), responsible for the transcription of protein-coding mRNA, as well as non-coding RNAs such as long noncoding RNAs and microRNAs, and Pol III-dependent tRNAs; 2) RiBi is a process deregulated in most, if not all, cancers and therefore Pol I transcription inhibitors have the potential to treat an extensive range of cancers; 3) the level of RiBi in healthy somatic cells is low, which makes these cells less sensitive to the effect of Pol I inhibition, thus increasing the selectivity of Pol I inhibitors towards malignant cells.

Here we review the current knowledge of the effect of the non-selective and selective inhibitors on rRNA transcription (Table 1). By analysing their mode of action and therapeutic potential we can gain invaluable information that will allow the development of new and even better inhibitors of Pol I transcription.

## 2. Drugs Targeting rDNA Transcription

A large variety of antineoplastic drugs are currently being used as therapeutics for cancer. These drugs can be classified accordingly to their mode of action into four major groups: Alkylating agents, that covalently attach alkyl groups to other molecules including DNA; Antimetabolites, that mimic the structures of normal metabolites; Antibiotics that bind/intercalate with DNA preventing RNA synthesis; and Plant alkaloids that prevent cell division [58]. Invariably these drugs have a range of efficiencies, tissue specificities and the totality of their molecular targets is unknown. Interestingly, recent studies [47] have demonstrated that several of these drugs directly or indirectly target transcription and/or processing of rRNA and suggest this property, likely at least in part, contributes to their therapeutic efficiency (Figure 3). Unfortunately, in such cases it has been difficult to precisely tease apart the contribution of RiBi inhibition to their therapeutic potential from their effects on other cellular processes and their toxicity. Importantly, however, more selective Pol I transcription inhibitors are being developed. These compounds are designed to selectively target the Pol I transcriptional machinery and thus it should be possible to determine the true potential Pol I inhibition for cancer therapy and to determine their toxicity profile.

### 2.1. Non-Selective Antineoplastic Drugs

**Alkylating agents**: Platinum-based alkylating compounds, like Cisplatin and its analogues Carboplatin and Oxaliplatin, are widely used chemotherapeutics in a variety of cancers, particularly those with high mortality rates like small cell lung carcinoma, ovarian and head and neck squamous cell carcinoma [59]. Platinum-based compounds react to form Platinum-DNA adducts. These adducts are detected by DNA repair proteins that become irreversibly bound to the DNA leading to cell death by mechanisms such as cell cycle arrest or inhibition of protein synthesis [60]. Although these drugs are often very efficient at clearing tumour cells, their use is associated with toxicity since it affects both cancer and normal cells (reviewed by Siddik [59]). Recent research has also shown that Platinum-based compounds, especially Oxaliplatin, also acts by inhibiting rRNA transcription and eliciting the NSP [47,48]. Cisplatin and Oxaliplatin are thought to inhibit rDNA transcription by displacement of the transcription factor UBF from the rDNA loci [49,50], resulting in the loss of recruitment of the PIC to the rDNA repeat and loss of the open chromatin state at the rDNA.

**Antimetabolites**: Antimetabolites are drugs that mimic the structure of cellular metabolites and inhibit specific enzymes causing DNA synthesis inhibition [58]. Several antimetabolite drugs are used as chemotherapeutics, including 5-fluorouracil (5-FU), an Uracil analogue (breast and colon cancer) and Methotrexate, a folate analogue (breast cancer, leukaemia) [61].

5-FU and its metabolites can be incorporated into RNA and DNA as well as bind to the active site of the Thymidine Synthetase enzyme, inhibiting its activity and consequently affecting the production of dTTP, leading to a deficit in Thymidine required for DNA synthesis. The combination of these attributes leads to both RNA and DNA damage and consequently the death of cancer or normal cells (reviewed by Longley et al. [62]). Thus, 5-FU affects rRNA biogenesis at the level of rRNA processing by blocking the maturation of the 47S [47].

Methotrexate can bind to the enzyme Dihydrofolate Reductase and prevents it from reducing folate. If folate is unavailable, the Thymidine Synthetase enzyme is incapable of converting Uracil into Thymidine leading to a deficit in Thymidine required for DNA synthesis. Burger et al. showed that treatment of cells with Methotrexate reduced Pol I transcription and leads to disruption of the nucleolus suggesting activation of NSP [47].

**Antibiotics**: Intercalating agents are antibiotics that exert their chemotherapeutic function by insertion between DNA bases. The downstream effects of the drugs in this category varies from induction of DNA damage (e.g., Actinomicyn D and Mitomycin C) and Topoisomerase poisoning (e.g., Doxorubicin and Mitoxantone). All these drugs have long been used as chemotherapeutic agent for the treatment of multiple types of cancer [61].

Actinomycin (ActD), also known as Dactinomycin, intercalates with both double and selective single stranded DNA, with a preference for GC rich sequences (reviewed by Longley et al. [63]). Intercalation of ActD ahead of the transcription fork causes RNA polymerases to pause and consequently blocks transcription elongation [64]. Not surprisingly, due to the level of rRNA in the cell and the high GC content of the rDNA, ActD preferentially inhibits Pol I transcription when compared to Pol II and III [51]. Intercalation of ActD into rDNA causes the formation of stable Topoisomerase I-DNA complexes preventing the progression of transcription and potentially leading to DNA damage [65]. Recent studies have also shown that ActD binds to G-quadruplex structures [66,67]. G-quadruplexes (G4) are four-stranded helical structures that can form in DNA and RNA due to the presence of four Guanines in close proximity. These structures have been identified in functional regions of the genome, like telomeres and promoter regions (for example in MYC promoter) and are predicted to have functional roles like protection of the telomeres and obstruction of DNA replication, transcription and translation [68]. Therefore, ActD may also interfere with transcription by stabilizing the G4 structures that occur in the regulatory elements of genes (including rDNA) leading to stalling of Pol I and consequently reduction of rDNA transcription. While low doses of ActD selectively inhibit Pol I in vitro, at the therapeutic doses used in humans it is not clear whether the therapeutic efficacy of ActD can be solely or partially attributed to selective Pol I inhibition or is a function of the inhibition of all three Polymerases.

Similarly to ActD, Mitomycin C (MMC), when in its reduced form, preferentially intercalates with GC-rich regions and alkylates DNA causing DNA damage [69] and ultimately cell death. MMC has been shown to preferentially inhibit rDNA transcription, most likely due to the high GC-content of rDNA [47].

Doxorubicin and Mitoxantone also intercalate with DNA but they act as Topoisomerase II poisons. Topoisomerase II enzymes remove DNA supercoils by a mechanism involving a transient break on both strands of DNA to allow strand passage. Intercalation with DNA leads to the formation of DNA adducts that prevent the Topoisomerase from re-ligating the DNA strands and therefore causes double strand breaks that cannot be repaired [52]. The DNA damage can both prevent DNA replication as well as induce cell death. Since Topoisomerase IIα plays an active role in PIC formation [39], it is no surprise that treatment of cells with either Doxorubicin or Mitoxantone inhibits rDNA transcription and leads to nucleolar disruption and activation of NSP.

**Plant Alkaloids:** Plant alkaloids are natural products that interfere with cell cycle and DNA synthesis [58]. One example of a plant alkaloid is Camptothecin, a natural product used in traditional Chinese medicine. Camptothecin was the first Topoisomerase inhibitor used in the clinic and other derivatives, including Topotecan and Irinotecan are also currently in use for the treatment of several cancers including, ovarian, small cell lung carcinoma and metastatic colon cancer [70].

Camptothecin, Topotecan and Irinotecan inhibit Type I Topoisomerases, that removes DNA supercoils by making a break in a single strand of DNA [71]. Topoisomerase has long been known to associate with Pol I [72] and interfere with rRNA transcription [53]. Inhibition of these enzymes leads to the formation of Topoisomerase I-DNA complexes that block transcription as well as increase DNA cleavage therefore promoting DNA damage [73].

Ellipticines are a family of planar plant alkaloids that can penetrate the cell by diffusion and inhibit Topoisomerase IIα. Several derivatives of the parent compound (e.g., hydroxyl-methyl-ellipticin and 9-hydroxy-ellipticin (9HE)) have been approved for use in clinical trials, but they have not progressed after stage one or two due to adverse side effects of the treatment [74]. However, as has been shown recently, ellipticines are also selective inhibitors of Pol I transcription initiation and cause dissociation of the SL-1 complex from the rRNA promoter both in vitro and in vivo. Similar to the selective Pol I inhibitor CX-5461, ellipticines activate NSP in a p53-dependent and p53-independent manner [54].

### 2.2. Selective Inhibitors of Pol I Transcription

The knowledge that Pol I transcription was frequently upregulated in cancer combined with the observation that many existing chemotherapeutic drugs possess substantive anti-Pol I transcription activity led to the realisation that drugs which selectively target Pol I transcription might constitute a new class of anticancer therapies with increased potency and reduced toxicity than non-selective inhibitors of RiBi. In early 2008, Cylene Pharmaceutical commenced high throughput screens for small molecules that could disrupt Nucleolin/rDNA G4 complexes and identified a series of fluoroquinolone derivatives, the lead compound of which was termed CX-3543 [55]. Subsequent to this, a second more selective and direct Pol I transcription inhibitor (CX-5461) was developed which demonstrated the ability to treat cancer in vivo [44]. By 2018, CX-5461 had completed a phase I clinical trials for haematological malignancies and is currently undergoing phase 1/2 trials for solid tumours. With this came a plethora of further studies and new compounds also targeting Pol I transcription, such as BMH21 [57] and the second-generation inhibitor PMR-116 being developed by Pimera Inc. While still early days, it is hoped that drugs targeting Pol I transcription have the potential to make a significant impact in the clinic for cancer treatment. We discuss the prominent selective Pol I inhibitors below.

**CX-3543**: CX-3543, also known as Quarfloxin (Figure 4A), was identified as a compound able to dissociate Nucleolin from putative G4 structures present at the rDNA locus leading to specific inhibition of rDNA transcription [55]. G4 are four-stranded helical structures that can be generated in single strands of DNA and RNA containing four or more consecutive guanines. The presence of these structures have implications for several essential cellular processes including transcription (reviewed by Rhodes and Lipps [68]). The presence of G4 at promoters, for example in the MYC gene, has been shown to inhibit transcription initiation [68]. However, it is hypothesized that G4 at the rDNA gene promotes transcription of rRNA [68]. G4 structures are present in the non-coding strand of rDNA which prevents re-annealing of the strands, therefore promoting higher transcription rates.

Nucleolin is a multifunctional RNA-binding protein with important roles in cell homeostasis and cancer. Nucleolin plays a role at several stages of RiBi: facilitates Pol I transcription by promoting the euchromatic state of the rDNA loci [75,76]; catalyses the cleavage of the 5′ ETS; and is involved in the assembly and transport of the ribosomal subunits [77]. Furthermore, Nucleolin is known to bind and stabilize G4 quadruplex structures, including the G4 quadruplex in the MYC promoter which leads to an inhibition of MYC expression. CX-3543 binds specifically and with high affinity to rDNA G4 structures causing dissociation of Nucleolin from these structures. This leads to a block in elongation and consequently a reduction in pre-rRNA levels. This inhibitory effect is selective for rRNA transcription since it does not affect Pol II-driven RNA transcription nor DNA and protein synthesis and has no effect on telomere function. The compound appears to specifically dissociate Nucleolin but no other important components of the Pol I transcription machinery (like UBF and SL-1), although an extensive analysis of rDNA interacting proteins has not been conducted. Inhibition of rRNA transcription by CX-3543 then leads to stabilization of p53 and induction of apoptosis via cleavage of Caspase 3. Treatment with this drug showed clear efficacy in growth inhibition over a range of cancer cell lines as well as in vivo anti-tumour activity [55].

CX-3543 has successful completed Phase I dose finding clinical trial [78] and reached Phase II clinical trial for Neuroendocrine and Carcinoid tumours (NCT00780663) but was withdrawn due to bioavailability issues [79].

**CX-5461**: CX-5461 (Figure 4B) was initially identified through a compound screen for selective inhibitors of Pol I transcription [56]. CX-5461 inhibits Pol I transcription at low concentrations in a range of cancer cell lines (IC50 <100mM) with a sensitivity of over 200 fold higher compared to Pol II inhibition. CX-5461 functions in part by inhibiting the formation of the PIC onto the rDNA template by specifically interfering with the interaction between SL-1 and Pol I (Figure 4A). In p53 wild-type cancers, CX-5461 induces apoptosis of cancer cells by activating the NSP [44] through nucleolar disruption, the sequestration of MDM2 by RPL5 and RPL11 and the consequent stabilization of the p53 protein and induction of apoptosis. In p53 null cancers, inhibition of CX-5461 can activate Ataxia-telangiectasia-mutated (ATM) and ataxia telangiectasia and Rad3-related (ATR)-mediated signalling in the absence of global DNA damage leading to arrest in cell cycle progression [80,81]. CX-5461 shows antiproliferative capacity in a broad range of cancer cell lines from different origins and with different p53 status [56,82]. Indeed, in solid tumours CX-5461 sensitivity does not correlate with p53 suggesting that pathways in addition to the NSP are important for its therapeutic efficacy. CX-5461 induces targeted DNA-damage in regions within and immediately adjacent to the nucleolus (RD Hannan et al., manuscript in preparation).

Another mechanism by which CX-5461 can lead to a reduction in tumour burden is by induction of differentiation. The levels of rDNA transcription are dynamic throughout haematopoiesis: haematopoietic stem cells (HSCs) have moderate levels of Pol I transcription that steadily increase through progressively lineage-restricted progenitors, while mature haematopoietic cells are characterized by low levels of rDNA transcription [83,84,85]. This observation is mostly justified by the need of HSCs and progenitors to tightly regulate protein synthesis [86]. Therefore, is not surprising that blast cells in haematological malignancy have the characteristics of haematopoietic progenitors and increased levels of Pol I transcription. We can therefore envisage that lowering rDNA transcription in the blasts to levels similar to that of mature cells, will induce terminal differentiation. Indeed, Hein et al. has shown that treatment of mice with acute myeloid leukaemia (AML) driven by the MLL/ENL fusion protein, reduces the number of myeloid blasts and increases the number of neutrophils in the bone marrow [82] suggesting that reducing Pol I transcription can directly affect haematopoietic differentiation. The mechanism by which reduced levels of Pol I transcription induces differentiation is unclear.

CX-5461 showed therapeutical efficiency in mice bearing MYC-driven lymphomas [44], models of p53 wild type (WT) and mutated murine AML, including MLL-driven leukaemias which are refractory to current therapies [82], and multiple myeloma [87]. Importantly, CX-5461 has also demonstrated therapeutic efficiency in patient-derived xenographs of AML and prostate cancer [82,88]. CX-5461 has also showed therapeutic potential in pre-clinical models of breast cancer [89], small cell lung cancer [90], ovarian cancer [91] and neuroblastoma [92] and in in vitro models of acute lymphoblastic leukaemia [81] and ovarian cancer (OVCAR4, RD Hannan et al., unpublished).

Interestingly, in more recent studies an additional mode of action for CX-5461 [89] was proposed where the drug intercalates into DNA sequences know to generate G4 quadruplex structures and stabilizes them. These structures are known to induce DNA damage leading to replication fork stalling and activation of DNA repair mechanisms, such as homologous recombination or non-homologous end joining. The authors use a fluorescence resonance energy transfer (FRET) melting assay and demonstrated an increase in melting temperature upon addition of 1 μM CX-5461. Of note, the concentration of CX-5461 used in this assay is known to mediate nonspecific activities, including inhibition of Pol II [56], and therefore the observed effect may be mediated by off-target effects. However, it is important to note that G4 structures are readily formed in vitro, also extrapolation to in vivo situations is not clear and therefore caution needs to be used when interpreting such results. Indeed, we have not found evidence that CX-5461 causes robust stabilization of G4 quadruplex DNA at doses that are therapeutically effective (Sornkom et al. under review). Moreover, in extensive high throughput DNA and RNA G-quadruplex binding assays conducted at the NCI, CX-5461 was not characterized as a G4 binder (Schneekloth et al., unpublished data).

CX-5461 has recently completed a Phase I clinical trial in patients with advanced haematological cancers [93]. In this first-in-human, phase I dose-escalation study, CX-5461 was determined safe and the maximum tolerated dose was identified. A decrease in rDNA transcription rates was observed after drug administration confirming on-target effect of CX-5461 in these patients. Clinical responses, including prolonged partial response and stable disease, were observed in approximately 30% of the patients. Clinical responses were observed in patients with both WT and mutated p53, with the majority of patients being p53 WT. However, the small number of patients in the study does not allow for an accurate prediction of p53 status to clinical response but shows a trend towards improved response in p53 WT. This would agree with known mechanism of action of CX-5461 in activation of NSR and stabilization of p53. Consistent with this, activation of the p53 pathway in samples from a patient post-CX-5461 administration were observed. This observation has important consequences for the design of future trials, at least in haematological patients, since p53 may be useful as both as a patient stratification tool and a biomarker for efficiency. It is also important to note that this Phase I clinical trial was performed in relapsed and refractory patients with haematological cancers that had failed all standard therapies and thus the clinical response to CX-5461 may be more pronounced in patients with less advance cancers.

CX5461 is also currently in Phase 1/2 clinical trial for breast cancer (NCT02719977). Similarly to the clinical trial in haematological cancers, CX-5461 was well tolerated and efficiency was observed in patients with homologous recombination-deficient tumours [94].

**BMH-21**: BMH-21 was the most potent of six compounds identified through a high content screen as being able to induce p53 activation. BMH-21 is a DNA intercalator however it does not activate the cellular DNA damage response [57]. It mediates growth inhibition at low concentrations in a range of cancer cell lines, but not in normal cells. The effective concentration varies between cell lines but is independent of p53 status. Since BMH-21 preferentially binds to GC-rich DNA, the effect of the compound on rDNA (64% GC) was investigated and it was shown to inhibit rDNA transcription, due to the disassembly of the Pol I complex at the rDNA promoter. The disassembly of the complex was due to the loss of one of its catalytic subunit RPA194. In the presence of BMH-21, RPA194 is dissociated from the Pol I complex and targeted by ubiquitin for degradation in the proteasome (Figure 4C). Interestingly, inhibition of ubiquitin-mediated proteasome degradation of RPA194 did not rescue Pol I transcription but lead to the accumulation of RPA194 at the nucleolar caps, the sites of stalled transcription of rDNA, suggesting that Pol I stalling is the primary effect of the treatment with BMH-21. This mechanism is different from the known DNA intercalator Actinomycin D, the Topoisomerase inhibitors and CX-5461 since it directly affects the Pol I complex by promoting the degradation of one of its components [95]. A recent report [96] proposed that impairment of Pol I transcription by BMH-21 activates a mammalian conserved regulatory checkpoint leading to targeting of RPA194 for degradation. The authors suggest that this occurs at the elongation stage of rDNA transcription based on in vitro assays and comparing the sensitivity of yeast mutants with defective elongation to wild type.

Two contradictory reports evaluated the potential of BMH-21 to intercalate and stabilize G4 DNA structures. Musso et al. [97] showed by nuclear magnetic resonance (NMR) and molecular modelling that BMH-21 binds the G4 found in the cMYC promoter and that this leads to downregulation of cMYC RNA with downstream effects on RiBi. Xu et al. [89], using a FRET melting assay, showed no effect of BMH-21 on the stability of G4. Direct evidence of the binding in vivo must be obtained to fully evaluate the role of BMH-21 in G4 stabilization. The observed G4 stabilization at the rDNA repeat may be a consequence of inhibition of Pol I transcription in general rather than due to the direct binding of the drug to the G4 structure. Indeed, the fact that CX-5461 and BMH-21 have very distinct chemical structures but have both been implicated in G4 binding suggests G4 stabilisation might be an indirect effect of these drugs.

### 2.3. Second Generation Inhibitors

The collective pre-clinical data on specific Pol I transcription inhibitors developed so far clearly demonstrate their therapeutic potential for the treatment of cancer. However, despite the clear benefits of these new drugs, CX-5461 is the only inhibitor of its class to have entered clinical trials (Australia and Canada, haematological and solid cancers). While pioneering and extremely promising, CX-5461 is associated with additional activities, like DNA damage (Hannan, unpublish data), which possibly contributes to its efficacy, toxicity profile and resistance mechanisms.

Therefore, the search for improved inhibitors or the second generation continues. A new chemical series of Pol I transcription inhibitors is currently under research by Pimera Inc. (personal communication). The lead molecule, PMR-116, shows improved chemical properties and higher efficiency compared to CX-5461 and is currently being prepared for first-in-human clinical trials with cancer patients. Further research is also being performed on other hits identified in the screen for BMH-21 [95].

## 3. Current Challenges

Although Pol I transcription inhibitor drug development is gathering pace, significant questions remain about their clinical application including patient stratification; what predicts resistance to Pol I transcription therapy; and what drugs might combine synergistically with Pol I transcription inhibitors to increase patient survival and overcome resistance. Finally, thought needs to be given to other steps in RiBi that might be amenable to drugging and whether the downstream signalling, and thus therapeutic effects, would be the same or differ to those Pol I transcription inhibitors currently developed.

### 3.1. Patient Stratification

The biggest challenges of cancer therapies that target housekeeping processes, like RiBi, is the identification of the types of cancer and groups patients that would best respond to the treatment. The use of the appropriate cohort of patients is crucial for the success of clinical trials. Testing on subjects not likely to respond will undermine the value of Pol I transcription inhibitors and potentially lead to reduced efficacy. Identification and validation of biomarkers is therefore a priority for the development of all new therapeutics, including Pol I transcription inhibitors.

Numerous publications have considered stratification of patients based on RiBi levels or nucleolus size [98] with mixed results, thus identification of a clinically relevant biomarker has remained elusive.

A potential biomarker for stratification of patients for Pol I transcription inhibition therapy is the presence of a functional p53 protein. Although NSP can also occur via a p53-independent pathway, the most common, and perhaps stronger, response to cellular stress is achieved by the p53- dependent NSP. Cellular stress leads to altered RiBi/Pol I transcription allowing for the stabilization of p53 and the activation of its downstream pathways leading to cell death or a block in cell proliferation [18,42,43]. Analysis of the response of haematopoietic malignancies to CX-5461 shows that p53 WT cancer cells are more sensitive, although mutant cells can still be responsive [44,80,82]. Indeed, results from the first-in-human clinical trial for CX-5461 strongly suggests that a functional p53 protein is a potential biomarker for treatment outcome. Although the number of patients in this study was limited, patients with WT p53 appear to respond better to the treatment [93]. In solid tumours, the response to Pol I transcription inhibition does not appear to correlate with p53 status, although increased sensitivity to Pol transcription inhibitors is also observed in some p53 WT tumours [56,88].

The expression level of the oncoprotein MYC (cMYC and N-MYC) is another potential biomarker to predict therapeutic response to Pol I transcription inhibitors. MYC expression is deregulated in a large proportion of cancers and its overexpression is linked with poor prognosis [99,100]. Through its function as a transcription factor MYC can directly regulate the expression of multiple genes involved in cell growth and division thus, impacting on a variety of processes including cell cycle regulation, protein synthesis and metabolism [101,102,103]. Recent experimental evidence supports a model whereby MYC is responsible for the global amplification of genes already expressed in normal cells rather than inducing new gene expression signatures [104,105]. Cancer cells become addicted to high levels of MYC to sustain their growth and therefore, are very sensitive to direct or indirect perturbation of MYC levels leading to proliferative arrest, differentiation or apoptosis [106]. MYC regulates RiBi by directly targeting protein-coding genes involved in RiBi, like UBF [13,107,108], RP and auxiliary proteins [109], as well as by activating Pol I- and Pol III- dependent transcription of 47S pre-rRNA and 5S rRNA respectively [83,107,108,110]. Consistent with the above, MYC-driven cancers have significantly increased levels of RiBi compared to non MYC-driven cells of the same lineage and are in general highly sensitive to Pol I transcription inhibitors [44]. One explanation for this heightened sensitivity of MYC-driven tumours to Pol I transcription inhibition is the ability of MYC to sensitise cells to NSP. In this paradigm MYC drives both rRNA synthesis and RP synthesis. Inhibition of rRNA synthesis by Pol I transcription inhibitors results in a massive excess of RPs, including RPL5 and RPL11 which in the absence of rRNA bind MDM2 and prevent the degradation of p53 leading to severe and prolonged activation of NSP and increased death of malignant cells compared to non MYC-driven malignancies (Figure 5). This is supported by the report that the level of p53 stabilization is dependent on the rate of ribosomal synthesis [111] and animal models of MYC-driven cancer respond well to treatment with CX-5461 [44,88]. Intriguingly, in MYC driven lymphomas, RiBi is in excess of requirements for the accelerated proliferation as rescuing apoptotic cell death induce by a 30–40% reduction by Pol I transcription inhibition restores a normal proliferative rate. This observation leads to the intriguing hypothesis that MYC has evolved to drive rRNA synthesis, not only to fuel growth and proliferation but to suppress p53 by ensuring RPs are consumed by rRNA during the processing stage of RiBi and are not available to bind MDM2 to suppress p53. This ensures that p53 is off when MYC is on and vice-versa.

Cancer cells without a functional homologous recombination mechanism, are known to be more sensitive to therapies that produce DNA double strand brakes, like radiotherapy and Topoisomerase II inhibitors [112]. Consequently, patients with mutations in genes like BRCA1 and BRCA2 may be particularly responsive to Pol I transcription inhibitors that induce DNA damage like CX-5461, but not to BMH-21 which do not appear to activate DNA damage.

The ATRX (alpha thalassemia/mental retardation X-linked) gene is mutated in approximately 5% of all cancers with the highest incidence in glioblastoma, neuroendocrine pancreatic cancer (53% with mutations in DAX/ATRX), non-small lung carcinoma and sarcoma [98,113,114]. ATRX is a chromatin remodeler that, in complex with the transcription factor DAXX, is essential for the deposition of histone H3.3 in repetitive regions, like the telomeres, centromeres and transposons [115]. Mutations in ATRX are associated with Alternative Lengthening of Telomeres (ALT), the process responsible for the maintenance of telomere [116] length in cancer cells thus allowing uncontrolled proliferation [117]. ATRX is also associated with the rDNA repeats and is responsible for the deposition of H3.3 and consequent trimethylation of histone H3K9 and H4K20, known markers of heterochromatin. Relevant here, the loss of ATRX leads to a decrease in the number of rDNA repeats and hyperactivation of the remaining repeats, probably reflecting a compensatory mechanism to maintain normal levels of rDNA transcription [116]. Intriguingly, tumour cells with ATRX loss and reduce rDNA repeat numbers demonstrate significantly increased sensitivity to the Pol I transcription inhibitor CX-5461. Possibly the hyperactivation of the reduced number of rDNA repeats renders cells unable to compensate in terms of Pol I transcription output in the presence of the Pol I transcription inhibitor accounting for the relative increased sensitivity. This observation predicts that ATRX mutant tumour cells might be more sensitive to Pol I transcription inhibition and therefore could be used for stratification of patients with cancer such as glioblastoma and neuroendocrine and prostate cancer that harbour ATRX mutations.

KDM4A (JMJD2A) is a member of the KDMA histone demethylase family and, among other targets (i.e., H3K36me3), it also removes the repressive chromatin mark H3K9me3 [118]. Therefore, its activity (at least in part) is opposite to that of ATRX. Recently, it has been shown that KDM4A is associated with active rDNA repeats and is required for efficient restoration of Pol I transcription in cells recovering from stress [25]. KDM4A is overexpressed in some cancers [119], particularly in prostate and breast cancer [120,121]. KDM4A expression is particularly high in triple negative breast cancer cells (TNBC) (Panov et al., unpublished data), and interestingly, the sensitivity of TNBC cells to CX-5461 and 9HE is inversely correlated with the level of KDM4A protein (Pearson coefficient from −0.89 to −0.98), while no correlation is observed for BMH-21 (Pearson coefficient −0.2) (Panov, unpublished data). This suggests that the expression level of KDM4A could be used to stratify TNBC patients for CX-5461 therapy.

### 3.2. Rational Identification of Combinatorial Therapies with Pol I Transcription Inhibitors to Reduce Relapse and Increase Survival

Pre-existing resistance and/or acquired resistance to therapy by circumvention of the drug targeted pathway is a common occurrence during cancer treatment. This is achieved by multiple mechanisms including pre-existing or acquired mutations in the direct drug target or downstream pathways or upregulation of alternative compensatory pathways. The use of a combination of drugs allows targeting of multiple pathways or one pathway at multiple levels, thereby reducing the chances of resistance development, is a common approach used to attain more durable responses in patients. Rational approaches to combinatorial therapies have been tested for Pol I transcription inhibitors with promising results [88,122].

The ability of MYC to drive protein synthesis is essential for cancer proliferation and can be inhibited at different levels from ribosome production to protein translation. Targeting protein synthesis at different levels is therefore predicted to improve therapeutic efficiency. Indeed, treatment of MYC-driven B-cell lymphoma with CX-5461, via inhibition of rDNA transcription, in combination with Phosphoinositide 3-kinase (PI3K), AKT and mammalian target of rapamycin complex (mTORC) inhibitors, disrupting the recruitment of ribosomes to the mRNA, lead to increased tumour cell death and prolonged survival [122]. However, certain types of cancer, like MYC-driven prostate cancer, are refractive to PI3K-AKT-mTORC inhibitors, but treatment with CX-5461 in combination with an inhibitor of PIM kinase, frequently upregulated in these cancers, showed improved therapeutic efficiency [88].

Inhibition of ATM/ATR pathways in murine models of AML [80] or poly (ADP-ribose) polymerase (PARP) in models of ovarian cancer (Sanij, Hannan et al., unpublished data) in combination with treatment with CX-5461 has also improved survival. A synergistic effect is achieved since disruption of the normal DNA damage response, i.e., prevention of DNA repair by the cancer cells, leads to increased cell death.

These results emphasize the fact that drug combinations need to be tailored to specific types of cancers and/or mutations. Therefore, several specific drug combinations are being assessed.

### 3.3. Rational Design of New Inhibitors

It is clear that selectively inhibiting Pol I transcription is an effective way to treat cancer without inducing general DNA damage and therefore would reduce toxicity and the probability of developing resistance. Pol I transcription can be inhibited at different stages and, because of the need of a multi-protein interaction to initiate and maintain Pol I transcription, a multitude of potential therapeutic targets can be envisioned. Fine mapping of the human Pol I complex and its associated proteins (e.g., SL-1 or RRN3) using crystallography or cryo-electron microscopy should allow us to design new inhibitors in a more targeted way, for example by fragment-based design. Using these techniques, we should be able to identify the protein interaction surfaces that would be more amenable to disruption by the presence of a small molecule. Together with the knowledge of the specific role of each of these proteins on Pol I transcription, i.e., knowing which proteins to target in order to obtain higher specificity and lower toxicity will be an invaluable asset. RRN3 provides one such example. RRN3, is necessary for the interaction between the SL-1 complex and PIC, mediated by the Pol I subunit RPA43. Studies demonstrate that disruption of the interaction between these complexes by CX-5461 leads to selective inhibition of Pol I transcription [56] suggesting that compounds targeting RRN3 directly might also have therapeutic efficacy. Consistent with this, a peptide mimicking the interaction region of RPA43 is able to compete for RRN3 binding and inhibit Pol I transcription [123] and studies using novel peptide deliver approaches of the RRN3 peptide for cancer treatment are already underway (L Rothblum, personal communication).

Perhaps one of the most attractive Pol I components for therapeutically targeting is UBF. UBF is a cytoarchitectural transcription factor that is essential for rDNA transcription. UBF binds at the rDNA promoter, across the Pol I transcribed portion of the rDNA repeat and remodels the chromatin at the rDNA locus to maintain permissive chromatin allowing for the fine-tuning of Pol I transcription levels in response to environmental changes [35,124,125]. UBF levels are directly related to the number of active rDNA repeats [35] and the number of UBF-bound active repeats are a feature of malignant transformation [126]. Therefore, drugs capable of interfering with the binding of UBF to the rDNA repeats would have good therapeutic efficacy. However it is relevant to note that UBF possess two isoforms due to alternative splicing of the same gene: UBF1 which binds specifically to the rDNA, and UBF2 which binds to many Pol II genes, in particular the histone clusters where its binding is essential for efficient Pol II transcription of both the canonical and noncanonical histones genes [124]. While targeting UBF1 would have therapeutic potential, the effectiveness of a pan-UBF1/2 inhibitor or a selective UBF2 inhibitor is less clear.

## 4. Conclusions

For many years, RiBi was considered exclusively a housekeeping process, and not a potential target for anti-cancer therapy. However, a paradigm shift has been observed over the past decade. It has become evident that selective inhibition of Pol I transcription has the potential to become a novel and efficient therapeutic approach in the fight against cancer. Moreover, it was convincingly demonstrated [47] that many of the existing anti-cancer drugs actually target various stages of RiBi and, in particular, rRNA synthesis. It has also become clear that inhibition of Pol I transcription has several important advantages as an approach to cancer treatment. Regardless of the heterogeneous nature of the cancer, the vast majority of cancer types rely on elevated levels of RiBi and, therefore, rRNA synthesis to support unrestrained proliferation. This makes Pol I transcription a “ubiquitous” target, present in almost every type of cancer. Furthermore, this target is highly specific, since Pol I (unlike Pol II and Pol III) transcribes only one gene, and due to the higher sensitivity of cancer cells to inhibition of RiBi, it is also highly selective towards malignant cells. All of this combined, potentially leads to less side effects and good complementarity of Pol I transcription inhibitors and existing therapies (e.g., radio- or immune-therapy). Importantly, that the combination of Pol I transcription inhibitors and other drugs (or treatment methods) will not only increase the effectiveness of the treatment as a whole, but also reduce the chances of developing acquired resistance.

## Figures and Tables

**Figure 1 cells-09-00266-f001:**
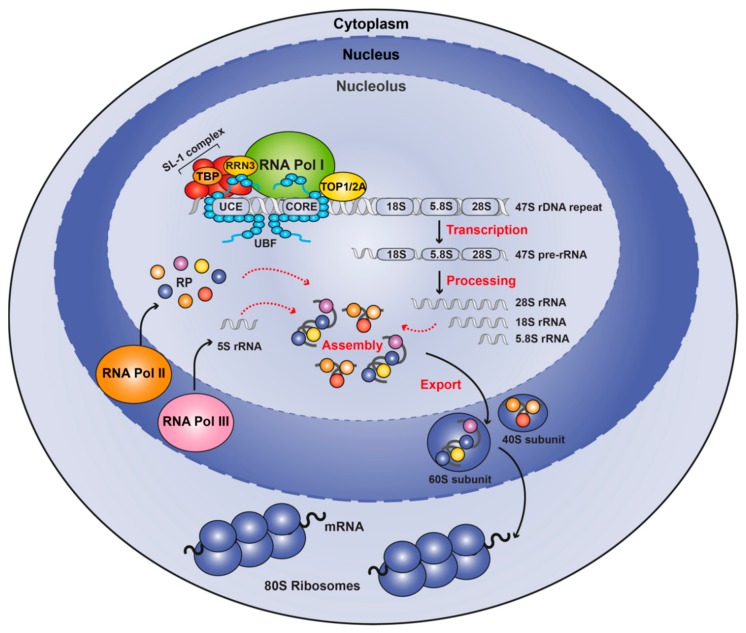
Schematic overview of ribosome biogenesis. Ribosome biogenesis occurs in nucleoli, which form around actively transcribing repeats of 47S ribosomal DNA. For Pol I transcription to initiate, UBF binds to the regulatory regions and recruits the SL-1 complex that interacts with Pol I via RRN3. Following synthesis, the 47S pre-rRNA is processed and modified to generate the mature 18S, 5.8S, and 28S rRNAs, which together with the 5S rRNA transcribed by Pol III, and ribosomal proteins transcribed by Pol II are assembled to form the 40S and 60S ribosomal subunits. These are subsequently exported from the nucleolus to the cytoplasm, where they form the mature 80S ribosome.

**Figure 2 cells-09-00266-f002:**
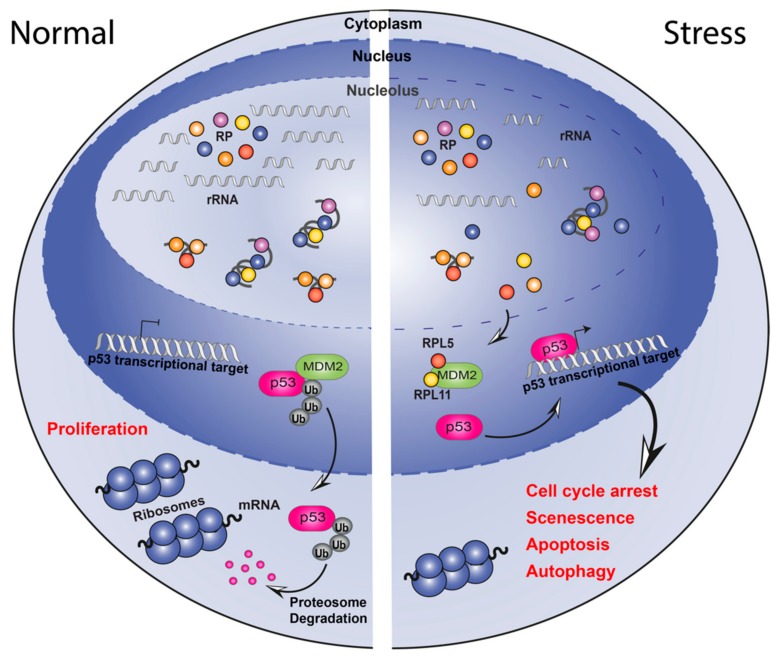
Schematic representation of the nucleolar surveillance pathway. Under normal growth conditions (left), ribosomal proteins (RPs) are assembled with ribosomal RNA into the 40S and 60S subunits in the nucleolus. p53 activity is maintained at low levels by ubiquitin-mediated degradation induced by MDM2. Following nucleolar stress (right), ribosome biogenesis is halted thus RPs, rRNAs, and other nucleolar proteins are released into the nucleoplasm. RPL5 and RPL11 are free to bind to MDM2 and prevent p53 ubiquitylation. p53 is then available for regulation of its target genes leading to cellular responses like cell cycle arrest, senescence, apoptosis and autophagy.

**Figure 3 cells-09-00266-f003:**
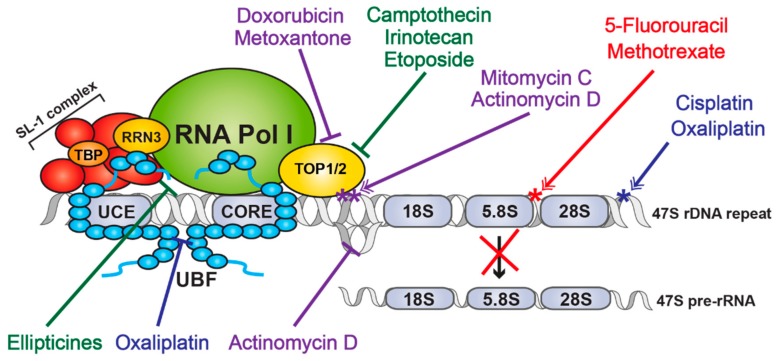
Schematic representation of mechanisms of action of unspecific Pol I transcription inhibitors. Figure showing an overview of the effects of unspecific Pol I transcription inhibitors. Blue - Alkylating agents; Red—Antimetabolites; Green—Plant alkaloids; Purple—Antibiotics. *—DNA adducts (purple), Platinum adducts (blue) or DNA damage (red).

**Figure 4 cells-09-00266-f004:**
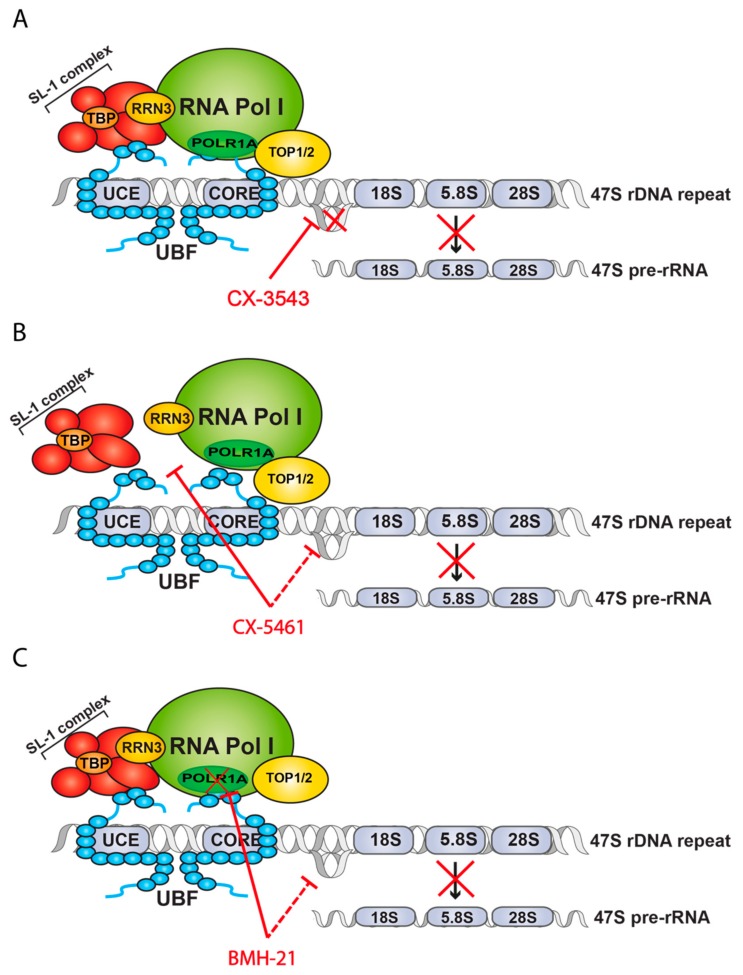
Schematic representation of mechanisms of action of specific Pol I transcription inhibitors. (**A**) Mode of action of CX-3543, disruption of Nucleolin binding to G4 DNA structures. (**B**) Mode of action of CX-5461, disruption of Pol I-SL-1 interaction and an unclear role in G4 stabilization. (**C**) Mode of action of BMH-21, disruption of Pol I complex and ubiquitin-mediated proteasome degradation of POLR1A (RPS194), inhibition of transcription elongation and an unclear role in G4 stabilization.

**Figure 5 cells-09-00266-f005:**
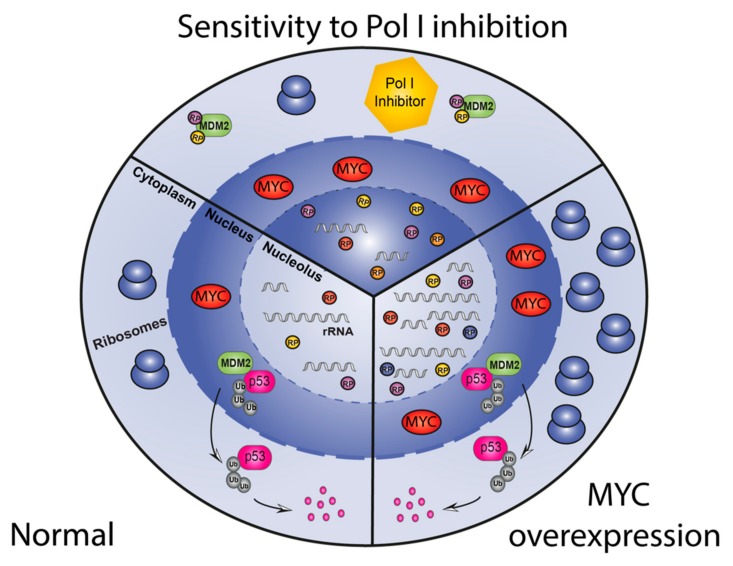
Schematic representation of the sensitization of MYC-overexpressing cancers to Pol I inhibition therapy. A cell with normal levels of MYC (left) maintains physiological levels of rRNA and RP and therefore produce adequate levels of ribosomes. Overexpression of MYC in cancer cells (right) causes increases of both rRNA and RP allowing for an increased number of ribosomes necessary to maintain uncontrolled growth and proliferation. Therapy with Pol I transcription inhibitors (top) leads to a reduction in rRNA levels while RP levels are maintained at a high level due to the effect of MYC on Pol II transcription. Excess free RP can then activate the NSP, allowing the stabilization of p53 and consequently promotes cell cycle arrest or apoptosis.

**Table 1 cells-09-00266-t001:** Chemotherapeutics affecting Pol I transcription.

Drug Type	Drug	Specific Pol I Inhibitor	General Mechanism of Action	Effect on Pol I Transcription	NSP Activation	Clinical Use	References
**Alkylating Agent**	Cisplatin	No	DNA damage DNA synthesis inhibition	Inhibition—Sequestration of UBF	Yes	Yes	[47,48,49]
	Oxaliplatin	No	DNA damage DNA synthesis inhibition	Inhibition—Sequestration of UBF	Yes	Yes	[47,49,50]
**Anti-Metabolite**	5-Fluorouracil	No	Uracil analogue Thymidine Synthetase inhibitor Intercalation into DNA and RNA	Inhibition of rRNA processing Intercalation into rRNA	No	Yes	[47]
	Methotoxate	No	Folate analogue Disruption of thymidine synthesis	Inhibition Undetermined mechanism	Yes	Yes	[47]
**Antibiotics**	Actinomycin D	No	DNA intercalation	Inhibition of transcription elongation	Yes	Yes	[51]
	Doxorubicin	No	DNA intercalation Topoisomerase II inhibitor	Inhibition—Likely prevention of transcription initiation	Yes	Yes	[52]
	Mitoxantrone	No	DNA intercalation Topoisomerase II inhibitor	Inhibition—Likely prevention of transcription initiation	Yes	Yes	[52]
	Mitomycin C	No	DNA alkylation	Inhibition—Undetermined mechanism	Yes	Yes	[47]
**Plant Alkaloids**	Campthotecin	No	Topoisomerase inhibitor DNA intercalation	Inhibition—Undetermined mechanism	Yes	Yes	[53]
	Irinotecan	No	Topoisomerase inhibitor	Inhibition—Undetermined mechanism	Yes	Yes	[47,53]
	Etoposide	No	Topoisomerase inhibitor	Inhibition—Undetermined mechanism	Yes	Yes	[53]
	Ellipticine derivatives	No	SL-1 displacement Topoisomerase II inhibitor	Inhibition of transcription initiation	Yes	Failed clinical trial phase II	[54]
**Specific Pol I Inhibitors**	CX-3543	No	Dissociation of Nucleolin-rDNA G-quadruplex complexes	Inhibition—Undetermined mechanism	Yes	Clinical trial phase II	[55]
	CX-5461	No	Disruption of interaction between SL-1 and Pol I at the rRNA promoter	Inhibition of transcription initiation	Yes	Clinical trial phase II	[56]
	BMH-21	No	Degradation of RPA194 and displacement of RRN3	Inhibition of transcription elongation	No	No	[57]

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
