# Peer review of "Targeting the RNA Polymerase I Transcription for Cancer Therapy Comes of Age"

_cells, 2020, doi:10.3390/cells9020266_

Round 1

Reviewer 1 Report

The authors have reviewed the biogenesis of ribosomes, the functioning of ribosomes in normal and cancer cells and the anticancer drugs targeting DNA polymerase I. The review is clear and well written. I have no major concerns about the manuscript.

To increase the novelty of the review and a more complete overview of the theme proposed in the manuscript, I suggest that the authors add a new paragraph describing the relationships among ribosome biology, polymerase I and the inflammation in cancer.

Author Response

We would like to thank the reviewer for their suggestion, however we feel the relationship between ribosome biology, RNA Polymerase I and inflammation in cancer is beyond the scope of this manuscript. There are limited publications addressing a possible relationship between targeting Pol I transcription, the focus for this review, and inflammation in cancer. Moreover, this is covered by an excellent review by Penzo et al (2019) in Cells.

Reviewer 2 Report

This is an excellent, well written review examining the potential of RNA polymerase I transcription inhibition as a cancer therapy by Ferreira and colleagues.  It provides a brief overview of ribosome biogenesis before focusing on RNA polymerase I and known therapeutic agents that target various components of the rRNA transcriptional machinery providing details of their mechanism(s) of action and clinical trial status.

One point which would help with the understanding the overview provided in section 1 of the review is that growth and proliferation are two separate but coupled processes in the cell.  Ribosome Biogenesis falls under cellular growth (increase in size) which is distinct from proliferation (increase in number). Fingar, D.C. & Blenis, J. Oncogene 23, 3151-3171 (2004) is one reference examining this.  There are also several examples of post mitotic cells that do not proliferate but their function can be altered by altering ribosome biogenesis substantiating the separate cellular functions of growth and proliferation.

Another important part of rDNA transcription is the participation of RNA helicases such as DDX5 which have been shown to be present at the rDNA promoter and require helicase activity to regulate rDNA transcription (Saporita et al. Cancer Research 17:6708 (2011)).  In addition, DDX21 has been shown to play a role as well (Calo et al. Nature 518:249 (2015). RNA helicases have also raised interest as potential therapeutic targets in cancer.  This is potentially a topic for an entire review on its own, but is another aspect of RNA Polymerase I regulation needed for a complete understanding.

Minor Points

1 - The first paragraph of section 1 (lines 33-51) would benefit from having references included for statements such as in line 44 "... not all tumors (<20%) are immune responsive..."

2 - Some of the text in Figure 1 is very small and difficult to read - such as the SL-1 complex members.

3 - Reference #102 is incomplete

Author Response

Reviewer comment 1: "One point which would help with the understanding the overview provided in section 1 of the review is that growth and proliferation are two separate but coupled processes in the cell." 

Reply: would like to thank the reviewer for the suggestion to further clarify the relationship between cell growth and cell proliferation and the role of ribosome biogenesis. We have included additional text and references in section 1 (page 2, lines 56-61) and throughout the text (highlighted in red) to clarify this matter.

Reviewer comment 1:"Another important part of rDNA transcription is the participation of RNA helicases"

Reply: We agree with the reviewer that the role of RNA helicases is emerging as an important regulator of the rDNA transcription and as such may be a therapeutic target for cancer. We also agree that to include the RNA helicases would make this review very lengthy. Thus, we have focused on the factors that are specific for Pol I transcription rather and not those that also have a function in Pol II or III transcription (Eg; DDX21). Furthermore we selected those factors which regulated Pol I transcription only and not processing (eg DDX5). Based on these guidelines to produce a focussed review we would consider that discussing the role of RNA helicases in beyond the scope of this manuscript.

Minor points

1 - The first paragraph of section 1 (lines 33-51) would benefit from having references included for statements such as in line 44 "... not all tumors (<20%) are immune responsive..."

Additional references (ref 1-5) were added to the first paragraph of section 1 (highlighted in red).

2 - Some of the text in Figure 1 is very small and difficult to read - such as the SL-1 complex members.

The letter size in Figure 1 has been increased and a new version of the figure added to the revised manuscript.

3 - Reference #102 is incomplete.

 Reference #102 has been corrected

Reviewer 3 Report

In this review, the authors present a nice overview of current status of targeting treatments for ribosomal biogenesis in cancer through Pol I transcription. The authors start with rationalization behind the idea of targeting such a fundamental mechanism like ribosomal biogenesis which is claimed to be shared among various cancer types. The review gives a good background on ribosomal biogenesis, with details on RNA Pol I complex, and how the associated mechanisms are altered in cancerous tissues. The manuscript then moves on the rationale behind targeting Pol I and reviews the current status of drugs specifically or non-specifically targeting Pol I. The authors conclude with an extensive discussion on challenges facing the treatments targeting Pol I, such as identifying biomarkers to determine the sensitivity levels.

Major Comments: Page 6: Authors list the advantages of targeting Pol I in cancer treatment. In doing this they refer to BRD4i. However, the comparison seems to require more in-depth explanation and analysis. BRD4 inhibitors are used to inhibit the epigenetic activation of super-enhancers and oncogenes globally. The prolonging of RNA POL II pausing is rather a consequential effect of BRD4i rather than targeted. The relation of RNA Pol II loading determines the efficacy of the BRD4i. This section requires some more clear explanation and recent citations related such as "Winter et al., 2017, Molecular Cell 67, 5–18" Page 6: Emphasis on the relation of Ribi and malignant cancer types would better emphasize the significance. Expanding on/adding a new section with specific mechanisms leading to Ribi alterations in specific cancer types would improve the understanding of the mechanism.  Such as which genomic alterations on ribosomal proteins or members of the POL I complex, are identified to cause Ribi upregulation? Adding the recent citations related to specific cancer types and sensitivity to Pol I targeted treatments and commenting on them might be helpful to put a perspective on specific cancer types. Low, J. Y., Sirajuddin, P., Moubarek, M., Agarwal, S., Rege, A., Guner, G., . . . Laiho, M. (2019). Effective targeting of RNA polymerase I in treatment-resistant prostate cancer. Prostate, 79(16), 1837-1851. doi: 10.1002/pros.23909 Cornelison, R., Dobbin, Z. C., Katre, A. A., Jeong, D. H., Zhang, Y., Chen, D., . . . Landen, C. N. (2017). Targeting RNA-Polymerase I in Both Chemosensitive and Chemoresistant Populations in Epithelial Ovarian Cancer. Clin Cancer Res, 23(21), 6529-6540. doi: 10.1158/1078-0432.CCR-17-0282 Minor Comments: Page 1, line 24, typographical error: “… progression, not only though increased protein synthesis and thus proliferative capacity but also” Page 2, line 82, typographical error: “… a subnuclear membraneless domains that facilitates the transcription of the rRNAs by Pol I and their” Page 3, line 116, typographical error: “… UBF also plays an important roles in promoter escape [30] and in regulation” Page 6, line 163: emphasis on “healthy somatic” should be made when referring to RiBi levels

Author Response

Reviewer comments 1: Authors list the advantages of targeting Pol I in cancer treatment. In doing this they refer to BRD4i. However, the comparison seems to require more in-depth explanation and analysis. BRD4 inhibitors are used to inhibit the epigenetic activation of super-enhancers and oncogenes globally. 

Reply: We understand the concerns of the reviewer regarding the lack of explanation regarding the use of BRD4 inhibitors. In this case we were referring to BRD4 and CDK9 inhibitors purely as examples of drugs targeting Pol II transcription. As such an in depth discussion of their mechanism of action we believe is not required. To make this clear we have changed the text on page 9 (lines 163-165: highlighted in red), from “target inhibition of Pol II-regulated genes (like BRD4 and CDK9 inhibitors)” to “target inhibition of Pol II-regulated genes by a variety of mechanisms (for example BRD4 [53] and CDK9 [54] inhibitors). We have also included references to recent review articles specific for these inhibitors for the reader.

Reviewer comments 2: Emphasis on the relation of Ribi and malignant cancer types would better emphasize the significance. Expanding on/adding a new section with specific mechanisms leading to Ribi alterations in specific cancer types would improve the understanding of the mechanism.  Such as which genomic alterations on ribosomal proteins or members of the POL I complex, are identified to cause Ribi upregulation? Adding the recent citations related to specific cancer types and sensitivity to Pol I targeted treatments and commenting on them might be helpful to put a perspective on specific cancer types.

Reply: We appreciate that specific changes in RiBi in different cancer types is of interest, however interestingly mutations in genes directly involved in the RNA Polymerase I transcription machinery are not common to any cancer type. In fact, there is more evidence that upregulation of Pol I transcription in cancer is mediated by a variety of upstream events, and it is these that are “cancer specific”.

Minor Comments: Page 1, line 24, typographical error: “… progression, not only though increased protein synthesis and thus proliferative capacity but also” Page 2, line 82, typographical error: “… a subnuclear membraneless domains that facilitates the transcription of the rRNAs by Pol I and their” Page 3, line 116, typographical error: “… UBF also plays an important roles in promoter escape [30] and in regulation” Page 6, line 163: emphasis on “healthy somatic” should be made when referring to RiBi levels

Reply: All the typographical errors detected by the reviewer have been corrected and are highlighted in red in the revised manuscript.